# Three-Dimensional Measurements of the Palate and Dental Arch Perimeter as Predictors for Maxillary Palatal Canine Impaction—A Cone-Beam Computed Tomography Image Analysis

**DOI:** 10.3390/diagnostics13101808

**Published:** 2023-05-20

**Authors:** Fadil Abdullah Kareem, Tara Ali Rasheed, Aras Maruf Rauf, Ranj Adil Jalal, Bestoon Mohammed Faraj

**Affiliations:** 1Department of POP, College of Dentistry, University of Sulaimani, Sulaimani 46001, Iraq; fadil.kareem@univsul.edu.iq (F.A.K.); tara.rasheed@univsul.edu.iq (T.A.R.); aras.rauf@univsul.edu.iq (A.M.R.); 2Department of Oral Diagnosis, College of Dentistry, University of Sulaimani, Sulaimani 46001, Iraq; ranj.jalal@univsul.edu.iq; 3Conservative Department, College of Dentistry, University of Sulaimani, Sulaimani 46001, Iraq

**Keywords:** palatal dimensions, arch perimeter, canine impaction, cone-beam computed tomography, MIMICS

## Abstract

(1) Background: Accurate diagnosis and treatment plans in orthodontics were facilitated by novel technologies. The prediction of occlusal problems is of utmost importance for clinicians. This present study aimed to find any possible correlation between unilateral and bilateral palatally impacted maxillary canine, palatal dimensions, and maxillary arch perimeter using digital measurements and determine the factors that could be used as predictors for maxillary palatal canine impaction. (2) Method: A cross-sectional study was conducted on cone-beam computed tomography images of orthodontic patients aged 15 to 25. Palatal dimensions and maxillary arch perimeter were digitally measured using Materialize Interactive Medical Image Control System. (3) Results: A significant difference was found between the case and the study groups regarding palatal depth, length, and arch perimeter, except for the palatal width. A weak correlation was reported in palatal depth and length with canine impaction, whereas the correlation was moderate concerning the arch perimeter. They can be used as predictors for palatal canine impaction. (4) Conclusion: Palatal dimensions and dental arch perimeter affect maxillary palatal canine impaction. Deficient arch perimeter, palatal depth, and length could be helpful in the prediction of maxillary palatal canine impaction.

## 1. Introduction 

The permanent maxillary canines are considered a cornerstone of occlusion and assume a critical job in smile design [1]. Moreover, they are among the last teeth to emerge in the oral cavity in the series of permanent maxillary normal eruptions [2]. After the maxillary first permanent molar, the maxillary canine is regarded as the most important tooth for dental occlusion due to its anatomy and location in the dental arch, and its protection is fundamental to its function. Keeping an individualized and proper inter-canine distance is also important for good esthetics [3].

For instance, in the Kurdish populace, the prevalence of unusually situated upper canines among subjects aged 12–22 years was 5.35%, and the most widely recognized issues included malposition (50%), trailed by canine rotation, displacement, impaction, and transposition [4]. Furthermore, in an Iraqi sample, the prevalence of canine impaction was higher (9.72%) [5]. The main difference between the terms impacted and retained is that in the first case, there is a physical barrier that prevents the correct eruption of the tooth, while in a retained molar, there is no evident impediment along its eruption path [6].

At the ages of 9–11 years, the position of the canine germ should be diagnosed for most children through clinical supervision. In seven to ten percent of children, the clinical investigation must be supplemented with intraoral radiographs in most cases [3]. A tooth is considered impacted if it’s unable to erupt after root completion or if its contralateral side erupts with complete root formation [7].

The upper canines are the teeth most frequently associated with impaction and failure of eruption. The long path of canines’ eruption, size, and timing of the eruption is extremely hard to decide [8]. It is important to diagnose cases of impaction early on and identify the etiological factors to achieve immediate and effective treatment per patient [9]. Then again, the frequency of buccally malposed maxillary canines is extremely high in the Iraqi populace.

The buccal impacted canine is mainly associated with crowding (lack of space in the arch), whereas palatal impaction may be due to the hypoplastic/missing lateral incisors (guidance theory) or with aplasia of molars and hypodontia (genetic theory) [10].

An impacted tooth is a totally or partially unerupted tooth, and its further eruption is improbable as it is located against another tooth, bone, or soft tissue [11]. Maxillary canine impaction is a widely observed orthodontic problem constituting a challenge to orthodontists to put them in their proper position and finish the case with a class I canine relationship [12]. If the displacement of the canines is detected early, the clinicians should focus on preventing a possible impaction [10].

Gupta et al., in 2012, reported that 85 and 15 percent of impacted permanent maxillary canines are palatal and labial impactions, respectively [13]. The reasons behind labial or buccal impaction are insufficient arch space and a vertical developmental position. When buccally impacted canines erupt, they emerge vertically, buccally, and higher in the alveolus. However, due to denser palatal bone, thicker palatal mucosa, and a more horizontal position, palatally displaced canines seldom erupt without complex orthodontic management [13]. One of the factors causing maxillary canine impaction could be a marked decrease in palatal depth. Thus, a study measured three palate dimensions, width, depth, and length, in upper canine impactions and compared them with cases of normally positioned canines via Cone-Beam Computed Tomography (CBCT) [14]. Transverse maxillary deficiency/or posterior cross-bite was also found to have a relation with canine impaction. Still, the study did not specify if the impactions were palatal or buccal. The prevalence of permanent tooth agenesis was higher in the cases of palatally displaced canines. Additionally, the mean mesiodistal width of maxillary lateral incisors in the palatally displaced canine was significantly smaller than in the normally erupted canine cases [15].

This current study investigated any possible correlation between three-dimensional palatal measurements and arch perimeter on one side and maxillary canine impaction on the other. This present study was conducted using the CBCT of patients with impacted upper canine to find out the correlation of palatal depth, dental arch dimensions, and perimeter with maxillary canine impaction in one study via the use of CBCT and Materialize Interactive Medical Image Control System (Mimics, Materialize Co., Leuven, Belgium) (MIMICS). To the authors’ knowledge, until now, there has been no other published research using the same evaluation method. This study’s findings may provide orthodontists with enough information regarding the prediction of canine impaction in an early stage of dentition.

## 2. Material and Methods

### 2.1. Methods

The ethical approval for this study was obtained by the Ethical Committee at the College of Dentistry, University of Sulaimani. A retrospective cross-sectional study of the CBCT for patients with unilateral and bilateral impacted maxillary canine (UIMC, BIMC) was conducted. Cone-beam tomography images (CBCT) were used to measure and confirm the impaction diagnosis. This study’s sample consisted of records from orthodontic patients who attended various orthodontic clinics in Sulaimani City/Iraq.

Out of 114 patients collected from different orthodontic clinics, only 60 patients (30 unilateral palatal canine impactions and 30 bilateral impactions regardless of the impacted canine’s position and orientation) who met the inclusion criteria were selected as a study group. All the selected cases had class I occlusion, as the frequency of canine impactions was significantly higher in class I malocclusion than in class II malocclusion [16]. The age of the patients ranged from 15 to 25 years.

In addition, it was reported that patients with skeletal Class III malocclusions did not show a different prevalence of canine impaction; therefore, it cannot be used as a risk factor for maxillary canine impaction [17].

In addition to the study group, another 30 CBCT images of fully erupted maxillary permanent teeth cases were selected as a control group.

### 2.2. Sample Size Estimation 

With a power of 95% and a significant level of 95% using G* Power 3.1 software, a sample size of 27 cases per group was calculated. Accordingly, the estimated sample size was 81 cases. The researchers recruited 90 cases, with 30 cases for each group to be more convenient.

### 2.3. Inclusion Criteria

Radiographs of patients 15–25 years of age with complete permanent teeth, excluding wisdom teeth.Cases with unilateral or bilateral impacted maxillary canine.No previous orthodontic treatment.No developmental anomalies, no history of trauma or craniofacial malformations.No intensive restorations or crowns.Class I molars dental relationship.Cases with high-quality CBCT images and volumetric data.

### 2.4. CBCT Imaging 

With the development of three-dimensional imaging techniques, CBCT has started to be applied in the diagnoses and treatment planning of impacted teeth. It also has numerous advantages; it is a predictable and reliable technique with lesser distortion, low cost, and requires lower doses of radiation than multiple bitewing radiographs [18]. 

In this present study, all the radiographs were taken by the same X-ray machine, which was a Pax-i3D machine (Vatech-Korea), with a field of view (FOV) of (100 × 85 mm), exposure time of 18 s, 94 KVp, and 8.4 mA. In this current study, all palatal measurements (width, depth, and length) and maxillary arch perimeter in both study and control cases were recorded and calculated from DICOM images by Materialize Interactive Medical Image Control System (MIMICS) software, using multiplanar reconstructions as well as the evaluation using 3D reconstructions. One examiner made all assessments, and the measurements were expressed in millimeters (mm). The palatal measurements and maxillary arch perimeter in this present study were analyzed as follows:

#### 2.4.1. Palatal Measurement

As illustrated in Figure 1A–C

-Palatal width was measured by drawing lines from the mesiobuccal cusp tip of the maxillary first molar from one side to the other [19]. The exact location of the buccal cusp tips of the upper first molar was identified and checked in all the planes from the coronal plane [20] and even from the reference plane. Then, the palatal width was measured from the axial view. Using the axial plane for the evaluation allowed the accurate designation of the selected landmark without superimposing different landmarks [20]. The coronal and axial levels at which the position of the mesiobuccal cusp of the upper molar was identified differ from one case to another due to anatomical variations, but in most of the cases, they are between (66–73 and 45–50) slices, respectively.The measurements were performed on an axial view of the radiographs derived from the CBCT images (Figure 1A).-Palatal depth is evaluated by the depth or height of the palate in the first molar areas [21]. The measurements were performed on a coronal slice of the radiographs derived from the CBCT images (Figure 1B).-The palatal length was determined by measuring the linear distance from the mesial contact point of the upper central incisors to the midpoint of the linear distance between the two upper first molars, obtained from the axial view of the radiographs derived from the CBCT images (Figure 1C).

#### 2.4.2. Maxillary Arch Perimeter

For calculating the maxillary arch perimeter, a transverse line connecting three points from the mesiobuccal cusp tip of the maxillary first molar to the mesiobuccal cusp tip of the opposite side, and the line then protracted from both sides anteriorly to the center point between the upper central incisors following the previous methodologies [19,21,22]. Calculations were performed on the axial view from the CBCT images (Figure 2).

### 2.5. Statistical Analysis

Descriptive and inferential analyses of collected data were performed through data entry to SPSS statistical software version 25. A *t*-test was applied to compare various parameters and canine impaction measurements. Correlation between parameters and canine impaction was found by the Pearson correlation method. A *p*-value ≤ 0.05 was regarded to be statistically significant. Finally, multiple regression analysis was carried out to consider all the factors simultaneously to determine significant associated factors with canine palatal impaction. 

## 3. Results

In this present study, 114 CBCT images were collected from different orthodontic clinics, and only 60 patients (30 UIMC, 30 BIMC), regardless of the impacted canine’s position and orientation who met the inclusion criteria, were selected as a study group. The age of the patients ranged from 15 to 25 years. In addition to the study group, another 30 CBCT images of fully erupted maxillary permanent teeth cases were selected as the control group without an impacted canine.

### 3.1. Descriptive Statistics

Means and standard deviations of parameters, including arch width, perimeter, palatal depth, and length, are all explained in Table 1.

### 3.2. Inferential Statistics

The inferential analysis of different parameters is shown in Table 2. A significant difference was found between the control and impaction groups regarding palatal depth (*p*-value 0.003), palatal length (*p*-value 0.000), and perimeter (*p*-value 0.000). The only recorded non-significant difference was inter-molar width (*p*-value 0.595).

The palatal vault was significantly more profound in the control group than in the unilateral and bilateral palatal impaction groups. A significant difference was found regarding the palatal length. A highly significant difference between the control and both impaction groups was reported concerning the arch perimeter.

The outcome about the presence of correlation of canine impaction with the various studied parameters, no correlation was found concerning inter-molar width (R = 0.106). In contrast, a weak correlation was recorded regarding each palatal depth (R = 0.308) and palatal length (R = 0.251), as well as a moderate correlation was found for the arch perimeter (R = 0.364). All details are demonstrated in Table 3. 

The multiple regression analysis results showed that each palatal depth, palatal length, and arch perimeter were significantly associated with palatally impacted canines, whereas inter-molar width was not significantly associated with palatally impacted canines, as shown in Table 4.

## 4. Discussion

After the third molar, the permanent maxillary canine is the most frequently impacted tooth. Buccally displaced impacted canines are more common than palatally displaced in Asian populations, while the opposite is true in Caucasian populations [3]. Many studies on maxillary canine impaction have tried to explain the causes of the impaction. As there are different etiological reasons for the impaction, the position of the impacted tooth is the most important variable that should be considered. Maxillary impacted canines (MIC), which have a significant place in orthodontics and can be easily noticed clinically, have frequently been a topic of research in the literature, along with the dentoalveolar and maxillofacial structures that are related to them [23]. As impaction assumes to be an approximately common phenomenon, orthodontists have to think about the frequency and the causative factors of impaction to assess appropriately any delay in eruption and diagnose early incidences of failure of eruption [9]. Some studies state a relationship between transverse width and impaction; others report no such relationship [24,25]. According to Siotou et al. 2022, the most common etiologic factor of canine impaction is the ectopic eruption path, followed by decreased intra-arch space and ankylosis of deciduous teeth [9].

Early detection and intervention of impactions are crucial to prevent further complications, such as root resorption, cyst formation, and esthetic problems. Correct diagnosis and treatment plans are important in managing impacted canines [26]. Prediction and treatment plans of impacted canines are challenging for orthodontists; many investigations were conducted to correlate this anomaly with craniomaxillofacial structures [24,27,28]. Three-dimensional palate and dental arch perimeter measurements can be used as predictors for maxillary palatal canine impaction [29]. However, this study investigated any possible relationship between three-dimensional palatal measurements and arch perimeter with maxillary canine impaction. Three-dimensional images showed higher efficacy in diagnosing and assessing impacted canines than other diagnostic tools.

Cone-beam computed tomography (CBCT) is a widely used imaging modality that provides accurate and detailed three-dimensional images of dentofacial structures. CBCT allows for precise measurements of the palatal and arch dimensions, which can aid in predicting the likelihood of palatal canine impaction [30]. This present study found that palatal depth and palatal length vary with canine impaction, which shows that canine development plays an important role in palatal configuration. Subsequently, the alterations in palatal dimensions could indicate canine impaction. A significant difference was found regarding the palatal length. A highly significant difference between the control and both impaction groups was reported concerning the arch perimeter.

According to Richardson, ref. [31] failure of the maxillary canine, which continues to erupt to move from the palatal side to the buccal side, may cause impaction. The multiple regression analysis results showed that each palatal depth, palatal length, and arch perimeter were significantly associated with palatally impacted canines, whereas inter-molar width was not significantly associated with palatally impacted canines.

Some studies also support our results regarding palatal depth, which show the significant difference between palatal depth and canine impaction [14,32]. In addition, a marked decrease in palatal depth was observed in impaction cases in a study by Omer and Alah in 2018, which stated that the result could be a factor in causing canine impaction14. In addition, our study accepts the results by Genc and Karaman in 2023, which revealed that palatal depths were lower in the impacted canine cases than in the control group [33].

On the contrary, Elmarhoumy 2021 [34] found that subjects with labial or palatal maxillary canine impaction do not show any statistically significant difference in the palatal vault depth compared with subjects without canine impaction. Meanwhile, based on a clinical study by Fathi et al. in 2012 [35], no significant correlation was noticed between palatal dimensions and canine impaction except for the arch length in the buccally impacted canine group, which was a statistically significant variable. Among the studied variables, the palatal width was not correlated with the factor of canine impaction, which agreed with the findings of some other researchers [14,34,36]. The likelihood of this finding could be explained by the position of the maxillary canines at the curvature of the dental arch when the anterior segment of the arch meets the posterior one. The absence of the canines will not necessarily alter the width of the palate with the presence of the rest of the teeth in normal occlusion. On the opposite side, some authors [37,38,39] stated that canine impaction is mainly associated with a deficiency in the maxillary transverse dimension; this controversy might be attributed to differences in the sample size, measurement protocol, and ethnic background of the subjects. In a study conducted by Cacciatore et al. 2018, they revealed that in the displaced maxillary canines group, the shape of the maxillary arch was narrower and shorter compared to the control group [38]. A study conducted by Maria et al. 2018 conducted that inter-molar width was not significantly reduced in subjects with buccally displaced canine, whereas the inter-canine width was significantly reduced, both at gingival and cuspidal levels. The difference may be due to the sample type and method of evaluation as they examined buccally displaced maxillary canines early in mixed dentition through inter-primary canine and inter-maxillary first molar on a digital cast [40].

In addition, the findings regarding arch width disagreed with Elmarhoumy 2021 who found smaller maxillary width in subjects with either a labial or palatal impacted canine than in subjects without impaction [34], while Yassaei et al. 2022 found that an impacted canine is significantly associated with a reduction in the mean maxillary arch width on the impacted canine side with no significant correlation between canine impaction and maxillary arch width [41].

Lombardo et al. [42] found that dental arches can be affected by ethnicity. They suggested that the upper and lower arch forms are more extended and broader in African individuals than those of Caucasians at the canines, premolars, and molars region. A maxillary arch perimeter was significantly reduced by canine impaction within the study group when compared to the control group; a similar phenomenon was reported by Al-Khateeb et al. in 2013 [32]. Moreover, Oleo et al. 2017 reported that approximately 80% of cases with palatally impacted teeth have a sufficient arch perimeter [43].

Dager et al. 2008 described the aging factor which influences dental arches. He concluded that dental arches change with age until the sixth decade of life. The changes in the arch width, length, and depth decrease with time. There is a tendency toward more rounded arch forms with age. This change in arch form results from a significant reduction in maxillary and mandibular arch depths [44]. The effect of canine impaction and the associated space loss cannot be neglected, especially in bilateral impacted cases.

However, Pasini et al., in a study conducted in 2020, suggested that the skeletal anomalies that are detectable in cephalometric radiographs at an early age can be a considerable sign for orthodontists during the diagnosis of maxillary canine impaction [45].

The absence of a tooth/teeth within the dental arch adversely affects the occlusion and dentoalveolar apparatus. Accordingly, in cases of an impacted canine, immediate and practical treatment is important for the orthodontic movement and preservation of impacted teeth in the dental arch [9]. Specific attention should be paid to the canines (the cornerstone of the dental arches) regarding the tooth size, time and path of eruption, orientation, and position. Orthodontic practitioners can utilize these predictors to develop personalized treatment plans for patients with maxillary palatal canine impaction, leading to improved outcomes and reduced complications. The limitations of this present study were the sample size and a single-time assessment of the parameters.

## 5. Conclusions

The discovery of the roles of this important tooth in the dental-maxillofacial region and the prediction of the undesired future upshots as a consequence of abnormal circumstances need rigorous investigation, to begin with the tooth germ development. Novel technologies facilitate the achievement of this goal. Maxillary palatal canine impaction in patients with Class I molar relationship causes alterations in palatal depth, palatal length, and maxillary arch perimeter to a limited extent, while palatal width remains unaffected.

More research should be encouraged to overcome the limitations of this present study with a larger sample size, considering the different prevalences of maxillary canine impaction in male and female patients.

Study limitations:

The authors did not evaluate the inter-canine width.

## Figures and Tables

**Figure 1 diagnostics-13-01808-f001:**
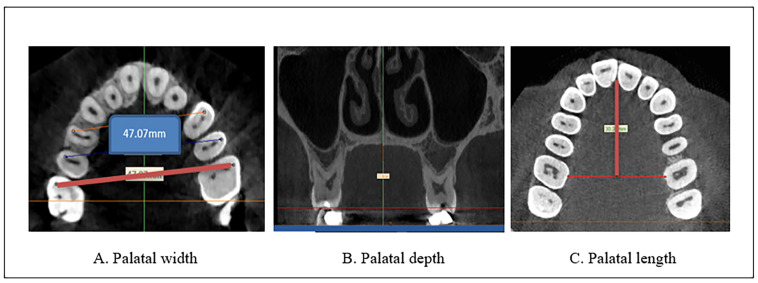
Palatal measurements with CBCT images using MIMICS software. (**A**) Palatal width (**B**) Palatal depth (**C**) Palatal length.

**Figure 2 diagnostics-13-01808-f002:**
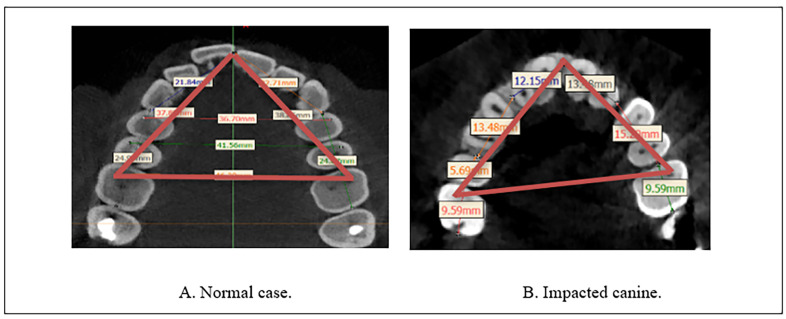
Calculating the arch perimeter from CBCT by MIMICS software. (**A**) Normal case (**B**) Impacted canine.

**Table 1 diagnostics-13-01808-t001:** Descriptive analysis of three-dimensional variables of the palate and dental arch perimeter.

Variables	No	Mean	SD
Palatal width	90	40.72	2.50
Palatal depth	90	20.68	2.44
Palatal length	90	29.64	3.35
Arch Perimeter	90	60.27	3.39

SD is standard deviation, No is number.

**Table 2 diagnostics-13-01808-t002:** F-test of canine condition with three-dimensional variables of the palate and the dental arch perimeter. Various variables of the study.

Parameters	Canine Condition	No	Mean	SD	F-Test	*p*-Value
Palatal width	Control/erupted	30	40.35	2.15	0.523	0.595
	Unilateral impacted	30	40.99	2.94		
	Bilateral impaction	30	40.84	2.38		
Palatal depth	Control/erupted	30	21.74	2.13	6.08	0.003 *
	Unilateral impacted	30	20.65	3.09		
	Bilateral impaction	30	19.65	1.40		
Palatal length	Control/erupted	30	30.82	1.33	30.06	0.000 *
	Unilateral impacted	30	26.64	3.54		
	Bilateral impaction	30	31.45	2.46		
Arch Perimeter	Control/erupted	30	62.01	3.21	11.09	0.000 *
	Unilateral impacted	30	58.31	3.20		
	Bilateral impaction	30	60.50	2.72		

SD is standard deviation, No is number. * significant.

**Table 3 diagnostics-13-01808-t003:** Pearson correlation of canine condition with various variables of the study.

Canine Condition		Palatal Width	Palatal Depth	Palatal Length	Arch Perimeter
Erupted or Impacted	Pearson correlation	0.106	−0.308 **	−0.251 *	−0.364 **
	R^2^	0.011	0.122	0.005	0.132
	Sig.	0.319	0.003	0.094	0.000
	N	90	90	90	90

** Correlation is significant at the 0.01 level (2-tailed). * Correlation is significant at the 0.05 level (2-tailed).

**Table 4 diagnostics-13-01808-t004:** Multiple regression analysis of canine status (as a dependent variable) and several co-variates (*n* = 4).

Model	Unstandardized Coefficients		Standardized Coefficients	*t*-Test	*p*-Value
	B	Std. Error	Beta		
**(Constant)**	**5.129**	2.036		2.520	0.014
**Palatal width**	0.026	0.032	0.080	0.817	0.416
**Palatal depth**	−0.130	0.033	−0.386	−3.922	0.000
**Palatal length**	0.046	0.025	0.187	1.859	0.067
**Perimeter**	−0.064	0.024	−0.265	−2.633	0.010

dependent variable (canine status: erupted, unilateral or bilateral palatal impaction).

## Data Availability

Data supporting reported results available on request.

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
