# Peer review of "Three-Dimensional Measurements of the Palate and Dental Arch Perimeter as Predictors for Maxillary Palatal Canine Impaction—A Cone-Beam Computed Tomography Image Analysis"

_diagnostics, 2023, doi:10.3390/diagnostics13101808_

Round 1

Reviewer 1 Report

The authors used Cone-beam tomography images to find possible parameters related to upper canine impaction. The novelty of this research was good. From the materials and method, the images of the CBCT and the image of the upper first molar buccal cusp tips seem not seem true cusp tips. How could we define the molar buccal cusp tip from the CBCT image? Because from my point of view, the images shown by authors may be under the cusp tip, not the real cusp tip.  Another question is why did the authors use class I occlusion patients but not class II or class III occlusion patients? From Angle's occlusion, the relationships between the three occlusions and the mandible relationships seem to be the major classification criteria. Does any research support that class II or III occlusions affect upper canine impaction? 

Author Response

Thank you, for all the respected editors and reviewers for their hard work and we tried to do all the corrections in the limited time.

Reviewer 1

  1. How could we define the molar buccal cusp tip from the CBCT image? DONE

  1. Another question is why did the authors use class I occlusion patients but not class II or class III occlusion patients? From Angle's occlusion, the relationships between the three occlusions and the mandible relationships seem to be the major classification criteria. Does any research support that class II or III occlusions affect upper canine impaction?  DONE

Reviewer 2 Report

Dear Authors,

The current study aimed to investigate any possible correlation between three-dimensional palatal measurements and arch perimeter on one side and maxillary canine impaction on the other.

In my opinion, the manuscript is suitable for publication in this Journal after major revisions.

Abstract

  • Abstract: The abstract should be a total of about 200 words maximum. The abstract should be a single paragraph and should follow the style of structured abstracts, but without headings: 1) Background: Place the question addressed in a broad context and highlight the purpose of the study; 2) Methods: Describe briefly the main methods or treatments applied. Include any relevant preregistration numbers, and species and strains of any animals used. 3) Results: Summarize the article's main findings; and 4) Conclusion: Indicate the main conclusions or interpretations. The abstract should be an objective representation of the article: it must not contain results which are not presented and substantiated in the main text and should not exaggerate the main conclusions.

https://www.mdpi.com/journal/diagnostics/instructions#preparation. Please follow the Instructions (structured abstracts, but without headings)

Introduction

I suggest organizing the introduction section as follow: after definition, please report epidemiological data “For instance, in a Kurdish populace, the prevalence of unusu-ally situated upper canines among subjects aged 12–22 years was 5.35%, and the most widely recognized issues included malposition (50%), trailed by canine rotation, displace-ment, impaction, and transposition .Furthermore, in an Iraqi sample, the prevalence of canine impaction was higher (9.72%).8” and then report etiology, and then how to diagnose impaction. 

Report the differences between retention and impaction.

Material and Methods

Did you include patients with Bolton discrepancy? Please better describe the inclusion and exclusion criteria.

Results

The Table 2 missed the abbreviations

Please report p-value as “P-value”

The values in the Tableas are not well positioned.

Discussion

This section was well organized and report recent literature on the topic.

References

The references should be reported according to the Instruction for Authors:

. Author 1, A.B.; Author 2, C.D. Title of the article. Abbreviated Journal Name YearVolume, page range.

https://www.mdpi.com/journal/diagnostics/instructions#preparation

Author Response

Thank you, for all the respected editors and reviewers for their hard work and we tried to do all the corrections in the limited time.

Reviewer 2

  1. The abstract should be a total of about 200 words maximum. The abstract should be a single paragraph and should follow the style of structured abstracts, but without headings. The abstract should be an objective representation of the article: it must not contain results which are not presented and substantiated in the main text and should not exaggerate the main conclusions.  DONE
  2. Introduction: DONE

  1. Material and Methods: Did you include patients with Bolton discrepancy? Please better describe the inclusion and exclusion criteria. DONE

Only we use the cases with molar class I angle relation without any regard to boltons discrepancy as we want to know if the 3 dimensions (not crowding) are related to the impaction.

  1. Results: The Table 2 missed the abbreviations, Please report p-value as “P-value”, The values in the Tables are not well positioned. DONE
  2. References: The references should be reported according to the Instruction for Authors. DONE

Reviewer 3 Report

Dear Authors,

The present study aimed to find any possible correlation between unilateral and bilateral palatally impacted maxillary canine, palatal dimensions, and maxillary arch perimeter using digital measurements and to determine the factors that could be used as predictors for maxillary palatal canine impaction.

The study was in line with the aims of the journal. 

However, there are some issues that should be addressed.

Abstract

Methods: A cross-sectional study was done on 90 cone-beam computed tomography images that were taken for patients as a part of the orthodontic diagnosis (60 cases with unilateral and bilateral palatally impacted maxillary canine versus 30 cases with normal occlusion).” Please, put “90”, “60”, etc in the result section and describe in this section only the methodology. 

I suggest to modify “normal occlusion” with “control group without impacted canine” or with a sentence that specify the clinical characteristics of these subjects (I dental class? No impaction? Etc)

Thus, report in the Results the number of recruited patients and their demographic data.

Introduction

Please report the differences between palatal and vestibular/buccal impaction, especially in terms of etiology.

What seems to be known is that buccal impacted are mainly associated with lack of space in the arch (crowding), whereas palatal with hypoplastic/missing lateral incisors (guidance theory) or with aplasia of molars and hypodontia (genetic theory) (Litsas, G. and Acar, A. (2011) A review of early displaced maxillary canines: etiology, diagnosis and interceptive treatment. The Open Dentistry Journal, 5, 39–47. Please report and discuss).

Methods

“Out of 114 patients collected from different orthodontic clinics, only 60 patients (30 unilateral palatal canine impactions and 30 bilateral impactions regardless of the impacted canine's position and orientation) who met the inclusion criteria were selected as a study group. The age of the patients ranged from 15-25 years. In addition to the study group, another 30 CBCT images of fully erupted maxillary permanent teeth cases were selected to act as a control group.” Please report all the numbers in the Result Section. 

Please report the inclusion criteria after “Sample size estimation”.

Results

The result section was clear and well organized. 

Number was reported as No in the Table one, and as N in Table 2.

Discussion

The authors did not evaluate the intercanine width. It should be added to the study limitation. 

The results showed that intermolar width was not significantly associated with palatally impacted canines. A study by the a research group conducted by Perillo (Association between 3D palatal morphology and upper arch dimensions in buccally displaced maxillary canines early in mixed dentition. Eur J Orthod. 2018 Nov 30;40(6):592-596. doi: 10.1093/ejo/cjy023. PMID: 29726936.) concluded that intermolar width was not significally reduced in subjects with buccally displaced canine, whereas the intercanine width was significantly reduce, both at gingival and at cuspidal levels. Please discuss these results in this Section.

Author Response

Thank you, for all the respected editors and reviewers for their hard work and we tried to do all the corrections in the limited time.

Reviewer 3

  1. Abstract: DONE
  2. Introduction; DONE
  3. Methods: DONE
  4. Results: DONE
  5. Discussion: DONE

Reviewer 4 Report

Please enter numbering of the lines, otherwise it becomes very difficult to make corrections 2. "who met the inclusion criteria": please mention the inclusion criteria first 3. “Out of 114 patients collected from different orthodontic clinics, only 60 patients (30 unilateral palatal canine impactions and 30 bilateral impactions regardless of the impacted canine's position and orientation) who met the inclusion criteria were selected as a study group. The age of the patients ranged from 15-25 years. In addition to the study group, another 30 CBCT images of fully erupted maxillary permanent teeth cases were selected to act as a control group. 2.2. Sample size estimation With a power of 95% and a significant level of 95% using G* Power 3.1 software, a sample size of 27 cases per group was calculated. Accordingly, the estimated sample size was 81 cases. The researchers recruited 90 cases, with 30 cases for each group. “ the explanation is not linear, the size of neither the control group nor the test group is clear. Please clarify 4. Tthe: correct 5. Palatal measurements: in measuring the length of the palate and in the width, specify in the axial view at which coronal or radicular level the measurement took place, in order to standardize it. In cases of significant molar torque, we can find several mm of difference between a coronal measurement and a more apical one 6. Same observation for all measurements. It is necessary to establish all the landmarks used. otherwise the system is questionable and not reproducible 7. Class I molars relationship: I find it very difficult that the patient in figure 2B has a 1st molar relationship on the affected side considering the considerable mesialization that is evident for the molar tooth. Clarify whether the I relation is molar or skeletal 8. What is intermolar? intermolar width... please do not change the terms used in the explanation and landmarks. It creates confusion 9. Table 2 clarify the abbreviations concerning the canin condition (R:) 10. -0.308** -0.251* -0.364** please clarify why they are presented once with - and once in positive value in the discussion. 11. Table 4 specify the abbreviations 12. Insert this usefull citation: Association between Anatomical Variations and maxillary canine impaction: A retrospective study in orthodontics Pasini, M., Giuca, M.R., Ligori, S., ...Marzo, G., Quinzi, V. Applied Sciences (Switzerland), 2020, 10(16), 5638

Author Response

Thank you, for all the respected editors and reviewers for their hard work and we tried to do all the corrections in the limited time.

Reviewer 4

  1. Please mention the inclusion criteria first. DONE
  2. Sample size estimation: the explanation is not linear, the size of neither the control group nor the test group is clear. DONE

The sample size after estimation was 27 for each group. To be more convenient, we increased the no. in each group to be 30.

  1. Title: DONE
  2. Palatal measurements: DONE
  3. Same observation for all measurements. DONE
  4. Class I molars relationship: Class I molar relationship which mean dental relationship. DONE
  5. What is intermolar? DONE
  6. Table 2 clarify the abbreviations concerning the canine condition DONE
  7. (R:) 10. -0.308** -0.251* -0.364** please clarify why they are presented once with - and once in positive value in the discussion. DONE

Correlation below zero express negative correlation.

  1. Table 4 specify the abbreviations DONE
  2. Insert this usefull citation: Association between Anatomical Variations and maxillary canine impaction: A retrospective study in orthodontics Pasini, M., Giuca, M.R., Ligori, S., ...Marzo, G., Quinzi, V. Applied Sciences (Switzerland), 2020, 10(16), 5638 DONE

Round 2

Reviewer 2 Report

Abstract

  • Abstract: The abstract should be a total of about 200 words maximum. The abstract should be a single paragraph and should follow the style of structured abstracts, but without headings: 1) Background: Place the question addressed in a broad context and highlight the purpose of the study; 2) Methods: Describe briefly the main methods or treatments applied. Include any relevant preregistration numbers, and species and strains of any animals used. 3) Results: Summarize the article's main findings; and 4) Conclusion: Indicate the main conclusions or interpretations. The abstract should be an objective representation of the article: it must not contain results which are not presented and substantiated in the main text and should not exaggerate the main conclusions.

https://www.mdpi.com/journal/diagnostics/instructions#preparation. Please follow the Instructions (structured abstracts, but without headings)

Author Response

Dear respected Editor and the reviewers
Thanks for your patience and your precious time, following, our responses (item to item) for each reviewer`s comments:
Reviewer 2 1. Abstract:
ï‚· The abstract should be a total of about 200 words maximum. Corrected accordingly ï‚· The abstract should be a single paragraph and should follow the style of structured abstracts, but without headings. Done
2. Introduction: The introduction was organized as suggested by the reviewer 3. Material and Methods: ï‚· Did you include patients with Bolton discrepancy? Bolton discrepancy was not considered ï‚· Please better describe the inclusion and exclusion criteria. Noted and considered
4. Results: ï‚· Table 2 missed the abbreviations. Corrected as requested ï‚· Report p-value as “P-value”. Done ï‚· The values in the Tableas are not well positioned. Done
5. Discussion: Thanks for the positive feedback
6. References:
ï‚· The references should be reported according to the Instruction for Authors All done

Reviewer 3 Report

Dear Authors,

The suggestions were not reported in the text. Please, modify the paper according to my previous comments.

Best regards

Author Response

Reviewer 3
The paper has been modified according to the reviewer`s comments from the first round

Reviewer 4 Report

The requested corrections have not been made.

1. template layout was not used and lines were not inserted, so it is not possible to mention the errors

2. In palatal measurements there are still several perplexities:

a. for the palatal palatal width you said that the measurement was taken from the tip of the mesio-buccal cusp, but the image with the measurement taken comes from a more apical section since you can see the pulp cavities therefore I have to deduce that all the measurements have been taken in the same way and therefore incorrectly,

b. for the palatal width you wrote a coronal slice which one ? at which mesio-distal level? At what level does the intermolar line pass?

c. "linear distance between the two upper first molars". where was it calculated from?"

d.in the maxillary arch perimeter same error: you said that the measurements were taken from the cusp tips and the photos show slices where you can see the pulp chambers which are obviously at more apical levels therefore I have to deduce that all the measurements taken are wrong

3. cit 40 is wrong there are all first names instead of surnames

4. I asked what was intermolar and was  you answered "done" but it is still there.

5. the tab. 4 is badly formatted and it is not clear what intermolar is already asked to correct

Author Response

Reviewer 4
1. Template layout was not used and lines were not inserted. Both points were considered
2. In palatal measurements there are still several perplexities
a. For the palatal width you said that the measurement was taken from the tip of the mesio-buccal cusp, but the image with the measurement taken comes from a more apical section since you can see the pulp cavities, therefore, I have to deduce that all the measurements have been taken in the same way and therefore incorrectly. As mentioned in the methods, the recorded measurements were taken from the tip of the cusps, these images, only, illustrate the way of measuring the distances for the readers. b. for the palatal width you wrote a coronal slice which one ? at which mesio-distal level? At what level does the intermolar line pass? c. "linear distance between the two upper first molars". where was it calculated from?" d. .in the maxillary arch perimeter same error: you said that the measurements were taken from the cusp tips and the photos show slices where you can see the pulp chambers which are obviously at more apical levels therefore I have to deduce that all the measurements taken are wrong. The images clarify the measurement methods, while the exact reference points were the tip of the cusp. 3. cit 40 is wrong there are all first names instead of surnames. Corrected 4. I asked what was intermolar and was you answered "done" but it is still there. 5. the tab. 4 is badly formatted and it is not clear what intermolar is already asked to correct Note: All the authors agreed and contributed evenly with the corrections Sincerely, The authors

Round 3

Reviewer 2 Report

Authors modified the text according to the suggestions.

I found this work impactful and it fits well with in the scope of this journal.

In my opinion, it is suitable for publication.

Reviewer 3 Report

The authors modified the text and the paper is suitable for publication.